# Palatine Tonsil Measurements and Echogenicity during Tonsillitis Using Ultrasonography: A Case–Control Study

**DOI:** 10.3390/diagnostics13040742

**Published:** 2023-02-15

**Authors:** Zohida A. Abdelgabar, Mahasin G. Hassan, Tasneem S. A. Elmahdi, Shanoo Sheikh, Wireen Leila T. Dator

**Affiliations:** 1Department of Radiological Sciences, Al-Ghad International Colleges for Applied Sciences, Madina 42541, Saudi Arabia; 2Department of Radiological Sciences, College of Health and Rehabilitation Sciences, Princess Nourah bint Abdulrahman University, P.O. Box 84428, Riyadh 11671, Saudi Arabia; 3Department of Medical Sciences, Al-Rayan College, P.O. Box 41411, Madina 42541, Saudi Arabia; 4Department of Health Sciences, College of Health and Rehabilitation Sciences, Princess Nourah bint Abdulrahman University, P.O. Box 84428, Riyadh 11671, Saudi Arabia; 5Department of Medical-Surgical Nursing, College of Nursing, Princess Nourah bint Abdulrahman University, P.O. Box 84428, Riyadh 11671, Saudi Arabia

**Keywords:** palatine tonsil, tonsillitis, ultrasonography

## Abstract

This case–control study aimed to assess the size and echogenicity of inflamed tonsils using ultrasonography. It was carried out at different hospitals, nurseries, and primary schools in Khartoum state. About 131 Sudanese volunteers between 1 and 24 years old were recruited. The sample included 79 volunteers with normal tonsils and 52 with tonsillitis according to hematological investigations. The sample was divided into groups according to age—1–5 years old, 6–10 years old, and more than ten years. Measurements in centimeters of height (AP) and width (transverse) of both tonsils (right and left) were taken. Echogenicity was assessed according to normal and abnormal appearances. A data collection sheet containing all the study variables was used. The independent samples test (*t*-test) showed an insignificant height difference between normal controls and cases with tonsillitis. The transverse diameter increased significantly with inflammation (*p*-value < 0.05) for both tonsils in all groups. Echogenicity can differentiate between normal and abnormal tonsils (*p*-value < 0.05 using the chi-square test) for samples from 1–5 years and 6–10 years. The study concluded that measurements and appearance are reliable indicators of tonsillitis, which can be confirmed with the use of ultrasonography, helping physicians to make the correct diagnosis and decisions.

## 1. Introduction

Enlargement of the tonsils is a frequent cause of apprehension for children and adults, as well as of doubts for general practitioners [1,2]. Although statistics show that tonsillitis only accounts for 1.3% of outpatient visits, poorly managed peritonsillar abscesses may advance adversely. This can lead to the development of fatal complications such as rheumatic fever, acute glomerulonephritis, and abscesses [3], as well as aspiration pneumonia, extension of the infection into deep neck spaces and the mediastinum, jugular vein thrombophlebitis, sepsis, and cavernous sinus thrombosis [4,5,6,7]. Many patients with recurrent tonsillitis experience hypertrophy of the tonsils that usually causes obstruction of the airway [8]. Often, the preoperative subjective tonsil sizes do not match with the actual mass, height, and width of the palatine tonsil in children and adults with tonsil hypertrophy and manifestations of obstructed airway clearance [9].

Tonsillectomy is commonly performed to relieve patients of obstructed airways—especially in children. The size of the tonsil can serve in calculating the success rate of tonsillectomy among children with enlarged tonsils. Significant for a successful surgical procedure for patients with obstructed airway clearance is a precise evaluation of the palatine tonsils [10].

On CT scanning, it is not easy to distinguish between anterior and posterior tonsillar pillars as their attenuation coefficients are similar. With T1-weighted MR Images, it is also challenging to separate the tonsils from the muscle because they are practically almost isointense. Meanwhile, with T2-weighted MR sequences, the tonsils present a higher signal intensity than muscles, due to the longer T2 relaxation time of the lymphoid tissue and submucosal glands within the tonsils comparative to the adjacent muscles [11].

Sans abnormalities, the tonsil normally surfaces as a well-defined hypoechoic structure with several echogenic reflections associated with the minimal presence of air. Differences in the size of the tonsils and inter-individual deviations are dependent on the age of individuals. With acute inflammation disorders, the tonsils are enlarged, with hypoechoic changes and unclear differentiation from the proximate tissue. An ultrasound can characterize a hypoechoic, visibly distinct space-occupying lesion touching the tonsillar bed; it displays the classic signs of inflammation and a central anechoic area—possibly with isolated internal echoes, suggesting cell fragments and distal acoustic enhancement [12].

During assessments of the throat, the form of the tonsil possibly provides a false approximation of its size. Several tonsils seem to rest considerably on the top of the throat, and others are deeply submerged much further in a deep tonsillar fossa [8]. Occasionally, the palatine tonsil is deeply seated and can hardly be felt superficially, despite being enlarged [13,14]. As an ultrasound may sometimes be a very useful diagnostic tool and as examinations can be performed when there is an infection or inflammation, this study aimed to characterize tonsils during inflammation, so that future ultrasound diagnoses can be more helpful and easier for children than laboratory investigations.

## 2. Materials and Methods

### 2.1. Sample and Area of the Study

About 131 Sudanese volunteers were recruited with the following eligibility criteria: between one and 24 years old; volunteers with normal tonsils and with acute tonsillitis; according to hematological investigations, a positive antistreptolysin O (ASO) titer test—this test is considered positive if the titer is more than 100 iU/mL for volunteers younger than 5 years and more than 200 iU/mL for volunteers older than 5 years. Those with chronic tonsillitis and peritonsillar abscess were excluded from the study. The sample was divided into three groups according to age: The first group included volunteers from 1–5 years, second from 6–10 years, and the third group included volunteers more than ten years old. Each group had normal cases as controls and cases with acute tonsillitis for comparison. The groups with no tonsillitis were classified as normal and those with acute tonsillitis as abnormal in this study. The study was carried out at different hospitals, nurseries, and primary schools in Khartoum state.

### 2.2. Study Design, Scanning, and Data Collection

This case–control study was approved by the institutional review board (IRB) at Sudan University for Sciences and Technology. For scanning a child, parental approval was taken before the procedure. For scanning an adult, their consent was required before the scanning. The data collection was carried out between November 2019 and October 2020.

Data were collected by scanning the tonsils of the patients with different portable ultrasound machines (ALOKA SSD-500, MINDARY, and ALOKA UST 5512U), and 5–10 MHz linear ultrasound transducers. Trans-cutaneous sonography was performed. The intraoral exam was avoided because it is commonly used in cases of peritonsillar abscesses. The volunteer was placed in a supine or sitting position with the neck extended during ultrasound scanning. The transducer was placed transversely below the mandibular angle. The tonsil was seen beneath the submandibular gland, above the constrictor muscle, and lateral to the tongue. In normal cases, the tonsil appeared well defined and in a hypoechoic form, with several echogenic reflections attributed to the presence of a little air, and the size of the tonsils was in accordance with the age of the individual. For inflamed tonsils, the palatine tonsils appeared as enlarged with variations in hypoechoic form and unclear differentiation from the proximate tissue. Two measurements in centimeters were taken: height (AP) and width (transverse) for both tonsils (RT and LT; see Figure 1). A data collection sheet was used to record age, history, sonographic measurements, and echogenicity. As the longitudinal and transverse diameters of tonsils are very examiner-dependent, and to ensure inter-rater reliability, two examiners performed all the ultrasound examinations. The percentage agreement reliability test was used to determine the agreement score.

### 2.3. Statistical Analysis

Statistical Package for the Social Sciences (IBM SPSS) program version 23 was used for the analysis. The frequency, percentage, and mean ± standard deviation (SD) were used to summarize the data. *p*-values and the comparison of the mean were based on the independent sample *t*-test and chi-square test. The comparison was considered significant if *p* < 0.05.

## 3. Results

### 3.1. Sample Distribution According to Age Groups

The study included 131 samples between one and 24 years old, divided into three groups according to age. Table 1 shows the sample distribution grouped according to age. Each group had individuals with acute tonsillitis, labelled as abnormal, and those without tonsillitis, labelled as normal. Out of 131 individuals, the first group with ages 1 to 5 years represented 48 (36%), with 18 (13.7%) being abnormal and 30 (22.9%) normal; the second group, with ages 6–10 years, represented 46 (35%) individuals, with 21 (16%) being abnormal and 25 (19%) normal; while the last group, aged older than 10 years, represented 37 (28%) individuals, of which 13 (9.9%) were abnormal and 24 (18.3%) were normal. The groups were 66 (50.4%) male and 65 (49.6%) female. The average weight was 28.2 kg with 10.7 SD, where 8 kg was the minimum and 50 kg the maximum.

### 3.2. Measurement of Tonsils

#### 3.2.1. Longitudinal and Transverse Measurement

The longitudinal (height) and transverse (width) of the tonsils from all groups were measured in cm. The differences in the Means ± SD of measurements for both tonsils in the normal and abnormal samples from all the age groups using an independent samples test (*t*-test) were calculated. The height (longitudinal) measurements of the right and left tonsils of all the samples with normal and abnormal tonsils from all age groups did not have significant differences, as shown by *p*-values > 0.05—whereas the width (transverse measurement) showed significant differences in the measurements, as shown by the *p*-values < 0.05 (Table 2).

#### 3.2.2. Echogenicity Measurements

Tonsil echogenicities were assessed for both normal and abnormal samples from the different age groups. The results were analyzed using the chi-square test (Table 3). The echogenicities of the right and left tonsils (RT, LT) of the normal and abnormal samples from the 1st (1–5 years old) and 2nd (6–10 years old) groups showed significant differences, as shown by *p*-values < 0.05. The difference was insignificant for the 3rd (>10 years old) group, as shown by *p* values > 0.05.

The test result variable(s): W has at least one tie between the positive actual state group and the negative actual state group (Figure 2; Table 4). Statistics may be biased.

A ROC was drawn for tonsillar width, which was significant among the first and second groups. The result showed a fair AUC (74%) [REF], CI [0.67–0.85]. The cut off value, which had a higher sensitivity equal to 57.1% and a higher specificity equal 85.5%, was a Tonsil width ≥ 1.51 cm (Table 5).

## 4. Discussion

This study was performed to determine the further promising value of ultrasound as one of the preferred diagnostic methods for determining conditions of the tonsil. Ultrasonography is not popularly sourced in prevailing clinical assessments of the oropharynx, but is being considered as an ensuring imaging modality for assessing the base of the tongue and the palatine tonsils [16,17,18,19,20,21]. Ultrasonography is comparable and complementary to Computerized Tomography (CT) and Magnetic Resonance Imaging (MRI). Although ultrasounds seem inferior, CT and MRI also have known critical drawbacks.

Computed tomography is ordinarily referred to as a CT scan. It is a diagnostic imaging procedure that employs a mix of X-rays and computer technology together to generate detailed images of the internal side of the body, including the bones, muscles, fat, organs, and blood vessels. It requires meticulous preparation and cooperation from the patient in order to complete the procedure [22]. Although it is able to provide much greater detail than ultrasonography, the scrupulous preparations before, during, and after the procedure makes it unbearable for many patients—especially among children and those with claustrophobia.

Aside from the painstaking procedure that prevents patients from undergoing CT scans, a high health risk is avoided from exposure to radiation. All X-rays emit ionizing radiation that has the capability to produce biological consequences in living tissues. CT and several nuclear medicine examinations are concomitant with far greater radiation doses compared to radiography. Specifically, the radiation amounts of various CT and nuclear medicine studies lie within a scale, shown by explicit epidemiological data, to be associated with increased risk of cancer [23]. Additionally, some patients may develop sensitivity reactions, and although rare, transient kidney failure due to the contrast agents used [24].

MRI, on the other hand, is a non-invasive imaging technology that yields three-dimensional detailed anatomical images. It is usually utilized to detect and diagnose diseases and check interventions. It is based on sophisticated technology that stimulates and traces changes in the course of the rotational axis of protons from the fluid that constitutes viable tissues. MRI scanners are remarkably well suited to image the soft tissues of the body [25]. MRI is deemed one of the best imaging modalities for children since, unlike CT, it does not have any ionizing radiation that could potentially be harmful. This makes it a better option for patients who require frequent imaging [26,27]; however, is also much more expensive than the CT scanner is.

In order to secure an MRI image, a patient is positioned in a huge magnet and instructed to be calm and not to make any single movement during the procedure to avoid blurs. The enclosed tube during imaging poses a health risk among people with claustrophobia. In addition, one of the most difficult challenges in MRI is letting a child or other patients keep still for a long period—hence requiring the use of anesthesia in order to get a clear image. Sedation exposes patients to additional health risks [28].

Contrast agents, which usually contain the element Gadolinium, may be prescribed to the patient intravenously before or during the MRI to raise the speed at which protons realign with the magnetic field. The image retrieved is brighter with faster proton realignment. Nevertheless, Gadolinium has some known adverse reactions, although minimal—the most common of which include headache, nausea, and dizziness for a short period post-contrast injection, and coldness at the injection site. Moreover, MRI uses a strong magnetic field and powerful forces that expand outside the machine and are intense enough to pitch a wheelchair beyond the room. Patients should notify their physicians of any form of medical implant prior to an MRI scan. It has been known that patients with iron-containing implants such as pacemakers, vagus nerve stimulators, implantable cardioverter- defibrillators, loop recorders, insulin pumps, cochlear implants, deep brain stimulators, and capsules from capsule endoscopy cannot undergo MRI [29]. With the advent of technology, this limitation has been addressed. Throughout the last six decades, tests for surgical titanium implants have been performed in several studies for their safety, compatibility, and imaging diagnostic artifacts. Almost all of the investigations reported that most nonferromagnetic implants are harmless for patients in MRI [30,31,32]. Another study also reported that most cochlear implants are now well-suited to magnetic resonance imaging (MRI) up to 3 T. Yet, despite this, the risk of critical mishaps is not absolutely disregarded. Reports on implant displacements and other adverse events with compatible implants have been reported in the literature [33].

MRI is known for its excellent accuracy in diagnosing neck problems, including tonsillar infections [34]. Several other limitations include MRI being costly and the waiting time being longer, which poses an increased risk of complications—while CT, which is also equally costly, unnecessarily exposes patient to radiation that predisposes the patient to cancer [35,36,37,38,39,40]. Ultrasonography clearly is the safer alternative. It has been repeatedly confirmed, despite the procedure being simple, that it is reliable. Related risks, if any, are incomparable to the risks from CT and MRI. It is accessible and much less expensive. In measuring palatine tonsils, patients—especially children—need not go through the intricacies of the CT and MRI to obtain reliable values—opting for ultrasonography is practical yet profound.

Among several studies, Ozturk, et al. (2017) reported measurements of Tonsil dimensions obtained with a 4.8–11.0 MHz transducer and required the patients to lie down and extend the neck in the opposite direction of the side being assessed; after which, the probe was placed below the lower chin and angled above the hyoid bone in both transverse and longitudinal planes, after clearly defining the mandibular angle. In this position, the tonsils were visualized as a well-defined hypoechoic structure lateral to the tongue root below the submandibular gland [41]. Ultrasonography is indeed straightforward and uncomplicated.

The measurements achieved through ultrasonography in this study were another valuable piece of evidence on the consistency and reliability of the capability of ultrasonography in measuring palatine tonsils. Patients are able to tolerate the procedure well as it does not require complicated preparations and actions before, during, and after, other than the proper positioning and movement of the neck. Tonsils are more dynamic in younger age and steadily diminish during adolescence [42]. The MRI studies performed by Aren [43] found that the palatine tonsil reaches a maximum size between seven and ten years and then gradually diminishes; because of this, the sample group in this study was divided into three groups, as there are different standard measurements for different ages. The significant result of this study from the various age groups, further classified into normal and abnormal, is indicative of the relevance of the use of ultrasound in the diagnosis of tonsillitis. Hosokawa, et al. (2020) discussed in their study that the use of ultrasound is not limited to diagnosing tonsillitis, but can also be used to determine the risk of sleep apnea [44].

The ultrasound results on the longitudinal and transverse diameter of the tonsils were considerable. Among all the groups, the study results showed that the longitudinal diameter of the tonsils did not increase consistently even in cases of tonsillitis and showed no differences to the tonsils of those without tonsillitis, as shown by *p*-values > 0.05. These findings suggest that the longitudinal diameter, which is also measurable using ultrasonography, may not be dependable for measuring the size of the tonsil.

The results also specify that the transverse diameter can be used confidently in estimating the size of the tonsil (*p*-value < 0.05 using independent sample *t*-test; see (Table 2). The study performed by Chikui [45] also confirms that the height of the node is not a trustworthy yardstick in the differential diagnosis of a cervical node; only the maximum transverse diameter of each node was used to determine the mean node size. Cui XW, et al. (2013) [46] noted that the short axis diameter increases specifically, and appears to be the paramount factor in small lymph nodes.

Concerning tonsillar echogenicity, it has been shown that it can be a predictive factor (*p*-value < 0.05 using chi- square test) for ages 1 to 10 years. As age advances, tonsils atrophy, and there is an interference in the echogenicity. The echogenicity of the tissue refers to “the ability to reflect or transmit US waves in the context of surrounding tissues” [47,48,49]. Several studies suggest the use of ultrasonography for determining tonsillar echogenicity, which is a reflection of the trust in its reliability. For cases other than tonsillitis that are considered to be complicated, requiring the use of either MRI or CT, ultrasonography is still considered a preliminary evaluation option. These results can be used for decision making and the determination of further medical action. A study compared diagnostic imaging and pathological analysis for space-occupying lesions of the parotid gland to establish the criteria for the differential diagnosis of benign and malignant types, with the use of MRI vs. powder Doppler. It concluded that both play a significant role in diagnosis and preoperative planning for parotid lesions. Ultrasound was considered for the first round of evaluation, revealing a description of the site, size of the lesion, and echo structure of the borders. The authors recommended the use of ultrasound over MRI due to the cost, accessibility, and patient’s acceptance of the technique [50].

The findings of this study further add to the confidence on the use of ultrasonography in the diagnosis of tonsillitis. Several other studies affirmed parallel results on the reliability of ultrasonography. Ultrasonography of the tonsils is a well-tolerated technique; besides being noninvasive, it is free from the risk of radiation [51,52] as well as sedation. Aydin and Uner (2020) also concluded in their study that ultrasonography could be used to diagnose palatine tonsil abnormalities [53]. A study conducted by Mengi, et al. (2020) used transcervical ultrasonography to measure palatine tonsil size and volume in both children and adults, and concluded that the use of ultrasonography is reliable for use in the objective measurement of palatine tonsil size and volume for children as well as adults [54]. Another study entitled “High-Frequency Ultrasound: A Novel Diagnostic Tool to Measure Pediatric Tonsils in 3 Dimensions,” claimed to be the first study to show that ultrasonography is a fitting empirical way to measure tonsillar volume in pediatric patients. Assessment of the tonsil structure and size prior to surgery can be carried out via ultrasonography to determine a risk stratification system for children with obstructive sleep apnea [55].

Ultrasonography is regarded as a dependable and effective method for isolating abscess development in the head and neck area, as well as measuring the palatine tonsils. A cohort study, where ultrasonography was used in 257 cases to determine the presence of abscesses and to delineate them from non-abscessing tonsillitis demonstrated an overall accuracy of 78.8% and concluded that abscess formation due to tonsillitis can be determined by ultrasonography, and therefore can be used as a first-line diagnostic tool to allow for prompt and suitable interventions [56]. In addition, although not usually used in diagnosing oropharyngeal disorders such as lymphoma [57], a case report on the Ultrasound features of primary non-Hodgkin’s lymphoma of the palatine tonsil concluded that ultrasonography can distinctly indicate the characteristics of primary lymphoma of the tonsils and could be a valuable imaging procedure in establishing oropharyngeal diseases. Although the authors strongly emphasized that a conclusive diagnosis can be ascertained only by histopathology [58], the potential of ultrasonography is diverse and evolving. In a pilot study carried out in 2019 on the use of 3D ultrasound for the accurate and quantitative measurement of the enlargement of tonsils, measurements were taken to determine the potential for the determination and classification of patients requiring adenotonsillectomy. The study revealed that the use of 3D ultrasound is feasible, but they were not confident whether the measurements obtained were accurate estimates and suggested further studies and the inclusion of larger sample sizes [59].

There may be other studies we have not found during our search that presented the potential of ultrasonography. We can confidently affirm based on our study and from several other authors that ultrasonography is no doubt accurate and reliable in measuring the transverse and longitudinal diameters of palatine tonsils.

## 5. Limitations of the Study

Cognizant of the different levels of expertise on ultrasonography that may alter the final image, the performing physicians in the study were well-trained and US-certified. The reports written by the ENT consultants were approved.

The use of different portable ultrasound machines (ALOKA SSD-500, MINDARY, and ALOKA UST 5512U) and 5–10 MHz linear ultrasound transducers in the data collection maybe considered a source of bias. These machines are used in the industry and are all considered reliable.

To address both possible sources of biases, an inter-rater reliability score using percent agreement between two raters was determined, with a yielded agreement of 92%. In general, 75% is acceptable, but 90% and above is the preferred level of agreement in clinical-related studies.

## 6. Conclusions

Ultrasound is a useful, noninvasive, and quick diagnostic tool—especially for children, who will be spared from radiation and sedation in cases of CT or MRI. It is a reliable and safe alternative to other prevalent diagnostic measurements for tonsillitis. It can be used to assess tonsils in cases of inflammation. Measurements and appearance are considered reliable indicators of tonsillitis. These objective measurements of the palatine tonsils with ultrasonography imaging serve as a reliable basis for either diagnosing palatine tonsil pathologies or making decisions on treatments, or both.

## Figures and Tables

**Figure 1 diagnostics-13-00742-f001:**
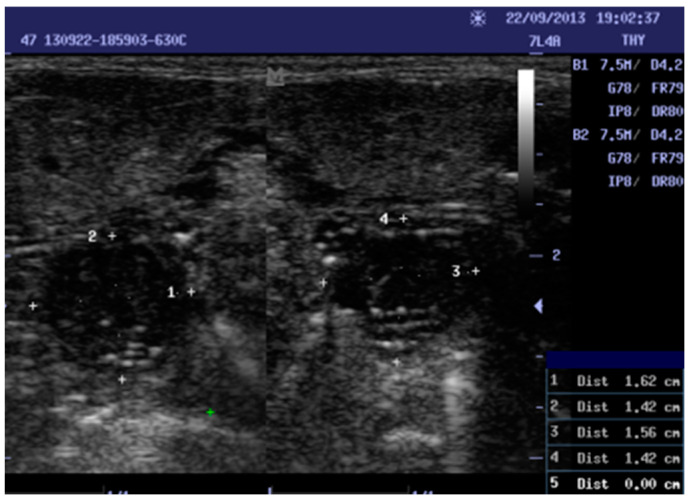
Shows a ten-year-old female with chronic tonsillitis 1, 3: width, 2, 4: Height. Right tonsil measurements (1.62 × 1.42). Left tonsil measurements (1.56 × 1.42).

**Figure 2 diagnostics-13-00742-f002:**
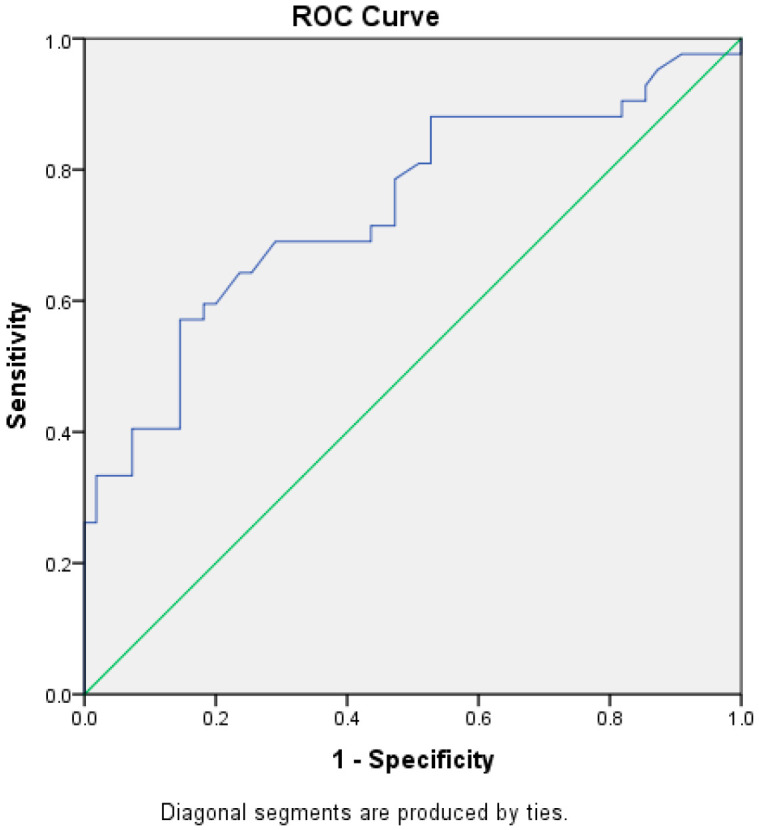
Receiver Operating Characteristic Curve (ROC) for the diagnostic ability of tonsillar transverse diameter.

**Table 1 diagnostics-13-00742-t001:** Frequency and percentage distribution of the sample according to age, gender, and weight (*n* = 131).

Group			Frequency	Percentage (within Groups)	Total Percentage
1st	1–5 years	Normal	30	62.5	22.9%
		Abnormal	18	37.5	13.7%
Subtotal			48	100	(36%)
2nd	6–10 years	Normal	25	54.3	19%
		Abnormal	21	45.7	16%
Subtotal			46		(35%)
3rd	>10 years	Normal	24	64.9	18.3%
		Abnormal	13	35.1	9.9%
Subtotal			37		(28%)
Total			131		100%
Gender		Male	66	50.4	
		Female	65	49.6	
Total			131	100%	
Weight		Minimum	Maximum	Mean (SD)	
		8 kg	50 kg	28.2 (10.7)	

Normal = no tonsillitis, abnormal = with acute tonsillitis.

**Table 2 diagnostics-13-00742-t002:** Measurement and *t*-test of differences between the height and width (in cm) of the normal and abnormal tonsils of all samples, categorized according to age.

Age Group	Tonsil	Measurement	Status	Mean ± SD	*t*	Sig. (2-Tailed)
1–5	RT	Longitudinal	Normal	1.4 ± 0.16	0.06	0.955
			Abnormal	1.4 ± 0.24		
		Transverse	Normal	1.3 ± 0.16	−3.5	**0.001 ***
			Abnormal	1.5 ± 0.22		
	LT	Longitudinal	Normal	1.4 ± 0.14	−0.22	0.820
			Abnormal	1.4 ± 0.18		
		Transverse	Normal	1.3 ± 0.16	−4.1	**0.000 ***
			Abnormal	1.5 ± 0.2		
6–10	RT	Longitudinal	Normal	1.5 ± 0.13	−0.657	0.514
			Abnormal	1.55 ± 0.22		
		Transverse	Normal	1.4 ± 0.15	−3.302	**0.002 ***
			Abnormal	1.6 ± 0.26		
	LT	Longitudinal	Normal	1.56 ± 0.12	0.408	0.685
			Abnormal	1.53 ± 0.20		
		Transverse	Normal	1.46 ± 0.14	−2.383	**0.022 ***
			Abnormal	1.6 ± 0.26		
>10	RT	Longitudinal	Normal	1.5 ± 0.15	−0.303	0.763
			Abnormal	1.5 ± 0.26		
		Transverse	Normal	1.4 ± 0.13	−3.327	**0.002 ***
			Abnormal	1.65 ± 0.26		
	LT	Longitudinal	Normal	1.57 ± 0.14	−0.256	0.799
			Abnormal	1.58 ± 0.29		
		Transverse	Normal	1.48 ± 0.11	−2.868	**0.007 ***
			Abnormal	1.67 ± 0.28		

Difference is significant if *p*-value < 0.05 *; normal = no tonsillitis, abnormal = with acute tonsilitis.

**Table 3 diagnostics-13-00742-t003:** Chi-square test on the differences between the echogenicity of normal and abnormal tonsils in the 1st group.

Age Group	Tonsil	Status	Echogenicities		*p*-Value
			Normal	Abnormal	
1–5	RT	Normal	30	12	0.001 *
		Abnormal	0	6	
	LT	Normal	30	12	0.001 *
		Abnormal	0	6	
6–10	RT	Normal	19	8	0.009 *
		Abnormal	6	13	
	LT	Normal	19	8	0.009 *
		Abnormal	6	13	
>10	RT	Normal	19	5	0.108
		Abnormal	7	6	
	LT	Normal	19	5	0.249
		Abnormal	8	5	

Difference is significant if *p*-value < 0.05 *; normal = no tonsilitis, abnormal = with tonsilitis.

**Table 4 diagnostics-13-00742-t004:** Area under the ROC Curve (AUC).

Area under the Curve				
Test Result Variable(s): W				
Area	Std. Error ^a^	Asymptotic Sig. ^b^	Asymptotic 95% Confidence Interval	
			Lower Bound	Upper Bound
0.743	0.052	0.000	0.640	0.845

^a^ Under the nonparametric assumption; ^b^ Null hypothesis: true area = 0.5.

**Table 5 diagnostics-13-00742-t005:** Optimal Cut-off point for the ROC Curve.

Positive If Greater than or Equal To ^a^	Sensitivity	1—Specificity	Specificity	Sensitivity + Specificity
1.51	0.571	0.145	0.855	1.426

REF: 0.90–1.00—Excellent; 0.80–0.89—good; 0.70–0.79—fair; 0.60–0.69—poor; 0.50–0.59—fail [15]. ^a^ ≥ 1.51 cm.

## Data Availability

The data that support the findings of this study are available from the corresponding author upon reasonable request.

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
