# Peer review of "Palatine Tonsil Measurements and Echogenicity during Tonsillitis Using Ultrasonography: A Case–Control Study"

_diagnostics, 2023, doi:10.3390/diagnostics13040742_

Round 1
Reviewer 1 Report
1. In line 32, you mentioned about peritonsillar abscess. Is peritonsillar abscess included in this study? They have very different symptoms and signs, different sonography features, and especially different treatment.
2. In line 69, for tonsillitis patients, are you enrolling acute cases or chronic cases?
3. In line 79, you need to clarify that you are using trans-cutaneous sonography.
4. In the result section, you will need to give more data about your patient demographics.
5. In table 1~3, what is your definition of normal and abnormal? Hemotalogical investigation and ASO titer is not specific to tonsillitis. Even if you want to use these measurements, you will need to inform your cutoff point.
6. In table 2, longitudinal and transverse length of tonsils is very examiner dependent. Total tonsil volume can be calculated by sonography exams, and will be much more scientific.
7. In line 130, for echogenicity measurements, mixed echogenicity is often noted in tonsils. How do you measure your echogenicity in such condition? You used three different sonography machines (ALOKA SSD-500), (MINDARY), (ALOKA UST 5512U), you will have to be sure that these machines measure the same echogenicity scores. Please arrange inter-rater reliability test.
8. In line 291, in the conclusion section, you mentioned “Measurements and appearance are considered reliable indicators of tonsillitis.” Please give us the cutoff point of each indicator you want to prove. Plot your ROC curve and tell us your AUC to defense your opinion.
Reviewer 2 Report
Dear authors, this is a relatively well presented study on the use of ultrasound for assessing tonsilitis in children. Indeed, this is a significant issue, as ultrasound can quickly assess a child in distress and provide an initial diagnsosis prior to ASO testing. However, a few points need clarification prior to publication. First, please go through and thoroughly check step by step the STROBE statement, and proceed to the appropriate adjustments, as some points are missing. Furthermore, the inclusion and exclusion criteria, as well as the overall selection process is not clarified (time period related or other). In addition, as it is well known, ultrasound is human-related, as different level of expertise can alter the final image. Even though the measurements taken are considered that of a basic level, a clarification should be made and a comment on the level of expertise of the performing physicians. Also, a comment on the current limitations of this study is a point that must be added. Finally, the discussion is adequate and the conclusions are carefully written.
Round 2
Reviewer 1 Report
We appreciate the author's effort for replying to our comments. I believe the work is much improved. There are some questions that I'm interested in:
1. In line 199~202, you talked about MRI-compatible implants. The statement is not true. Most implants, including cochlear implants, are now compatible to MRI. You may want to clarify that to avoid misunderstanding.
2. For point 8, what is "greater than or equal to 1.51"? Do you mean largest tonsil width under sonography? Then you might want to change the description in table as "Tonsil width ≥ 1.51 cm". You may consider adding this part to the article.
